# Examining the Disruptive Potential of Generation Z Tourists on the Travel Industry in the Digital Age

**Alina Petronela Pricope Vancia** *[ID], **Codruța Adina Băltescu** [ID], **Gabriel Brătucu** [ID], **Alina Simona Tecău,**
**Ioana Bianca Chițu** [ID] and **Liliana Duguleană** [ID]

Faculty of Economic Sciences and Business Administration, Transilvania University of Brașov, Colina Universității Street No. 1, Building A, 500068 Brașov, Romania; codruta.baltescu@unitbv.ro (C.A.B.); gabriel.bratucu@unitbv.ro (G.B.); alina_tecau@unitbv.ro (A.S.T.); ioana.chitu@unitbv.ro (I.B.C.); ldugul@unitbv.ro (L.D.)

*  Correspondence: alina.pricope@unitbv.ro

**Abstract:** The tourism industry has faced several challenges over the years, due to the evolution of technology and behavioral changes of the generations. The research focused on the new generation of tourists, Generation Z. Known as digital natives, the study aimed to identify their current travel behavior in the digital age and their perception of the future of travel in the context of recent technological developments, namely artificial intelligence, and virtual reality, thus highlighting specific elements that could disrupt the travel industry. To achieve this goal, qualitative research was conducted, using two sessions of focus groups among 20 Generation Z tourists. The results show that Generation Z tourists exhibit disruptive behavior primarily due to their heavy reliance on social media platforms, even for travel purposes. Social media has become their primary search engine, and travel influencers hold significant sway over certain individuals in this generation. In addition, they serve as influencers, by sharing visual content from their travels. Another noteworthy trend in the travel industry is Generation Z's inclination towards multi-channel booking, effortlessly switching between different booking options. Lastly, although the results show little awareness of the potential of advanced technologies, their openness to adopt them to simplify the travel planning process further contributes to the disruption of traditional travel patterns. Generation Z can be considered a bridge between previous and future generations. The study has implications for management and marketing activities in the tourism field.

**Keywords:** Generation Z; tourism disruptors; distribution channels; travel influencer; artificial intelligence; virtual reality

## 1. Introduction

According to most researchers and media sources [1–4], Generation Z includes those born between the mid to late 1990s and the early 2010s, and they are considered the first unique and truly digital natives [2,5], with the internet being omnipresent in their lives [6]. Generation Z is often described as self-loving, determined, and ambitious, exhibiting preparedness and caution [7]. They have a realistic outlook, are open to dialogue, and possess a high social tolerance. Their consumption values revolve around seeking unique, infinite, and ethical products [4,8]. Generation Z values customization and personalization, expecting tailored experiences that cater to their individual needs and preferences [9–11]. Furthermore, they are influenced by major events, such as climate change, war, and terrorism, showing concern for social issues beyond their own self-interests and engaging with global matters [7].

Generation Z is considered one of the most interesting generations in terms of the future of tourism due to the characteristics it present [12]. Adults of Generation Z are, at the time of writing, the new generation of tourists, to whom tourism businesses should

pay great attention. Studies show that tourists from this generation have a significant impact on the tourism industry [8] since they are digital natives and use technology at all stages of their journey [13]. Generation Z has attracted many researchers in the field of tourism on topics such as their needs, their behavior as tourists, [14–16] and their concern for sustainable consumption at tourist destinations [17–20].

Studies of the behavior of the new generation of tourists have preoccupied researchers worldwide. Setiawan et al. [21], for example, demonstrated the digital tourist quality of Generation Z in South Jakarta. Their research showed that Generation Z members use online media rather than conventional travel agencies to search for information and make reservations. The places they visit need to be "Instagrammable" as they share photos and videos from their trips on their social media accounts. Another study conducted among Generation Z in the United States [22] examined the sustainable behavior of Generation Z as their digital natives on the choice of transport modes. The results of this research show that Generation Z, being significantly influenced by technological advances, could have great potential to support sustainable transport. Other studies have sought to find similarities and differences in tourist behavior between them and other generations. These have included members from countries such as Italy [23], Serbia [24], and New Zealand [20].

More recently, several studies have captured the behavior of Generation Z regarding the potential for developing virtual tourism [25], as well as the role of mixed reality in cultural heritage tourism and assessing digital natives' awareness of all the capabilities of these advanced technologies [26]. Participants in these studies came from countries such as Poland, in the first study case article, and Kazakhstan, India, Kyrgyzstan, Uzbekistan, Russia, Vietnam, Thailand, Cambodia, and Azerbaijan in the second study case article.

However, the authors consider that the study of the behavior of this generation as emerging tourists needs further research as their correct understanding is essential for stakeholders in the industry to develop effective tourism policies and strategies for the future, especially in the context of the emergence of advanced technologies.

Given the numerous research studies conducted on the relationship between tourism in the digital age and the behavior of Generation Z in different countries, as shown above, it is noteworthy that in Romania, to the authors' knowledge, an exploration of this correlation is missing. Thus far, there was interest in discovering, among Generation Z tourists in Romania, their favorite destinations and the tourist activities they prefer during their trips [27], as well as their sustainable behavior at the destinations [17,28]. Therefore, conducting a research study in order to explore the behavior of Romanian Generation Z tourists in the context of digitalization and the adoption of new technologies was considered essential to fill this gap in the literature. Through it, specific information on this demographic group in Romania can be acquired, in order to complete previous studies and to be used in other scientific research studies for comparison with different nationalities. The significance of conducting research in this area is highlighted by Romania's recognition as a key player in outbound tourism within Europe. The country has experienced a considerable number of departures over the years, exemplified by the substantial figure of 17,265.3 thousand departures recorded in 2022 [29].

With this paper, the authors seek to draw attention to adult travelers of Generation Z in Romania, aiming to identify their current travel behavior in the digital age and their perception of the future of travel in the context of recent technological developments, namely artificial intelligence, and virtual reality, highlighting elements that disrupt or may disrupt the tourism industry. To this end, an exploratory survey was conducted to gather information from Generation Z members from Romania who had undertaken at least one leisure trip within the 12 months prior to the research. Participants in the study were interviewed using the group interview method.

The study was designed to be exploratory; the authors did not formulate any hypotheses beforehand. The focus of this research was based on finding answers to the following questions:

→　Q1: How do Generation Z organize their trips?

$\rightarrow$ Q1.1: What are the main factors influencing the vacation purchase decision of Generation Z?

$\rightarrow$ Q1.2: What are the sources of information used by the members of Generation Z regarding travel?

$\rightarrow$ Q1.3: What are the methods used by Generation Z to book their trips?

$\rightarrow$ Q2: What is the utility of social networks during the travels of Generation Z members?

$\rightarrow$ Q3: Can advanced technologies (artificial intelligence and virtual reality) influence Generation Z's future travel behavior?

Answering the above questions could help identify potential disruptive behaviors and opinions toward the travel industry.

After introducing the context, the paper presents the literature highlighting the influence and importance of Generation Z tourists, as well as the dynamics of the travel industry in the digital age. Then, in sequence, the methodology, results, and discussion of the research are presented. The final section of the paper provides conclusions and related implications.

## 2. Literature Review

### 2.1. Generational Shifts in Tourism: The Influence and Significance of Young Travellers

People's expectations, requirements, or preferences regarding the purpose, place, and time of their holiday vary depending on their age [30]. The tourism industry has evolved with each generation, adding new dynamics [31–33]. Understanding how generational changes affect tourist behavior can help predict and adapt to future tourism trends effectively [33].

Currently, there exist four identifiable generations whose age permits them to engage in travel activities [30]. Those are Baby Boomers (born between 1940 and 1959), Generation X (born between 1960 and 1979), Generation Y (born between 1980 and 1994), and Generation Z (born after 1995) [4]. Members of those generations exhibit specific behaviors and consumption patterns due to the context in which they were born and lived most of their lives.

In the post-World War II era, Baby boomers emerged as a generation deeply influenced by their consumption habits as a means of expressing their ideology. Generation X, in contrast, focused on consuming goods and services as symbols of social status, while Millennials leaned towards consuming experiences as a way to shape their identity. For Generation Z, the driving force behind their consumption choices lies in the pursuit of truth, both individually and within their communities. This generation comfortably embraces the idea of not adhering to a single way of self-expression. Their quest for authenticity not only grants them greater freedom to express themselves but also fosters a remarkable openness and empathy towards diverse individuals, facilitating a broader understanding of different perspectives [4].

The inherent variations among generations have naturally resulted in unique travel behaviors [34]. A study by Expedia Group Media Solutions [35] provides insights into the travel behavior of the four generations as follows. Baby boomers, characterized by their confidence and decisive nature, prioritize active exploration, sightseeing, and culinary experiences. Generation X, often seeking family-oriented trips, values cultural experiences, and enjoys off-the-beaten-path activities and recommendations. Millennials, known for their frequent travel and YOLO (you only live once) mindset [36], prioritize diverse experiences, outdoor exploration, and budget-consciousness. They share their experiences online and are accompanied by Generation Alpha, their digitally native children. Generation Z, the mobile-first generation, prioritizes relaxation, bucket-list activities, and cultural experiences. They heavily rely on social media for travel inspiration and use mobile devices for research and booking.

In addition to recognizing the significance of all generations in the tourism ecosystem, the travel industry acknowledges the importance of young travelers [37]. Therefore, it can be argued that Generation Z currently plays a crucial role in shaping the future of the

tourism industry [38,39]. Dimitriou and AbouElgheit [40] even proposed a modified model specifically adapted to the travel purchase decision-making process for this generation as a result of a comprehensive analysis of previous research on their travel behavior. Their suggested model comprises five key stages: inspiration, desire for social recognition, planning, search, evaluation, booking, and post-booking evaluation. This model diverges, for example, from the traditional Engel–Kollat–Blackwell (EKB) model of consumer decision-making, which encompasses the following stages: problem recognition, search, alternative evaluation purchase, choice, and outcomes [41].

### 2.2. Dynamics of the Travel Industry in the Digital Age

Advances in technology, information, and communications in recent decades have led to rapid changes, which in turn have revolutionized the entire tourism sector [42]. Innovative approaches, modern technologies, and authentic experiences tailored to tourists' needs foster satisfaction and loyalty toward destinations, as highlighted by recent studies [43]. Moreover, tourists have proven to adapt easily to these technologies, as the convenience of using them has outweighed the potential risks, such as being tracked while traveling [44].

Consequently, technological advancements have caused significant disruption across every aspect of the tourism industry [45], from the emergence of global distribution systems [46–48] to advanced technologies such as artificial intelligence and the Metaverse [49].

Online travel agencies (OTAs) are one of the most impressive examples of digital transformation of distribution channels in the tourism sector [50] and are still widely used today. Brands such as Booking.com and Expedia are just two of the best-known companies of this kind [51], which attract hundreds of millions of visitors worldwide to their websites every month [52]. In the same category, another industry disruptor is meta-search platforms. These search engines use technology to allow web visitors to search multiple online platforms simultaneously and help them find more results without having to search multiple web browsers [53]. The best-known platform of this kind is TripAdvisor, which uses the pay-per-click advertising business model as its distribution model [54].

Social media has changed the way people search, find, read, gather, share, develop, and consume information, and the way they communicate with each other and share information [55]. The popularity of this phenomenon has once again led to changes in the tourism industry [56,57]. Tourism businesses have become increasingly concerned about their presence on social media platforms as it allows them to interact directly with clients and constantly monitor customer opinions and evaluations of their services [58]. Moreover, in the digital age, visual representation in social media is more impactful than written text in terms of expressing personal experiences [59,60]. In the context of tourism, travelers can easily use their mobile phones to photograph or film the places they visit on vacation, posting them on social media platforms, thus generating considerable increases in consumption and content creation [61,62].

Mobile technologies are also playing a role in changes in the tourism industry [63,64]. It gives privileges to consumers to identify, customize, and purchase tourism products and services, and suppliers by providing tools to develop, manage, and distribute their offerings [65]. Tourists not only use mobile technology to help them plan their trip, but they also use it during their vacations to communicate, navigate to find activities, relieve boredom, and take photos [66].

The sharing economy is also a controversial topic in the tourism industry [67,68]. Platforms, such as Airbnb and Uber, have disrupted the traditional travel industry by offering alternative accommodation and transport options [69–72]. These peer-to-peer platforms have enabled travelers to find unique accommodations, local experiences, and transportation services, instead of hotel accommodations and traditional transporters [73].

More recently, themes of advanced technologies as future disruptors in the travel and tourism industry have appeared in the literature. These include Blockchain, by providing decentralized and transparent solutions for various actions such as payment processing and loyalty programs [74–77], artificial intelligence, by using chatbots, virtual assistants,

and recommendation engines [78–80] and Metaverse [49], by the use of advanced reality technologies such as augmented reality, virtual reality, mixed reality, and other emerging technologies that will allow experiences in ways not possible in the real world [81].

The travel industry can maintain its relevance and enhance customer experiences by acknowledging and embracing both generational changes and technological advancements. Generation Z's profound connection to the realm of technology is inherently strong, primarily because they are recognized as digital natives. As individuals born into a world saturated with digital advancements, they possess a natural affinity and deep familiarity with various technological tools and platforms. This generation effortlessly integrates technology into their daily lives, relying on it for communication, information access, entertainment, and much more. Their inborn understanding of digital technologies and seamless adoption of emerging innovations solidify the profound association between Generation Z and the ever-expanding sphere of technology.

## 3. Materials and Methods

Based on the research questions and the review of existing literature, the research objectives were established as follows: (1) to explore the main elements that influence Generation Z's decision-making process when purchasing vacations, (2) to investigate the main sources of information used by Generation Z regarding travel, (3) to identify the distribution channels used by Generation Z to purchase vacations, (4) to evaluate the importance of social media during travel for Generation Z, and (5) to assess the extent to which advanced technologies may influence the future travel behavior of Generation Z.

To obtain primary information from some members of Generation Z, we considered that the most appropriate qualitative research method to achieve the objectives set for this study was the focus group discussion. This method is based on the theory of dynamics of groups according to which a small group is a collection of interdependent individuals between whom there are intense interactions [82]. The characteristics of Generation Z, detailed at the beginning of this paper, in particular, their openness to dialogue and their tolerance towards the opinions of others, also contributed to the choice of the focus group interview, as it allows interaction on several levels—both between respondents and between researcher and respondents [83], thus adding value to the research.

To collect extensive narratives and descriptions on the study subject, criteria were established beforehand to select the participants, which included: (1) being between 18 and 26 years old (adults of Generation Z), (2) having taken at least one trip in the last 12 months prior to the time of the study, (3) being familiar with concepts specific to advanced technologies (artificial intelligence, virtual reality, mixed reality, Metaverse), and (4) not having participated in a similar research study in the last 6 months. It was essential to find attendees who met these criteria, and for this reason, the recruitment of participants was carried out in a location with a potentially high concentration of Generation Z members, such as a university environment. This process resulted in two group interview sessions, with 20 respondents, 10 in each. The demographic characteristics of the respondents are presented in Table 1.

**Table 1.** The demographic characteristics of the respondents.

| Variable | Frequency | Percentage |
|---|---|---|
| **Gender** | | |
| Female | 12 | 60 |
| Male | 8 | 40 |
| **Age** | | |
| 18–22 | 16 | 80 |
| 23–26 | 4 | 20 |
| **Marital Status** | | |
| Single | 7 | 35 |
| In a relationship | 13 | 65 |

Source: the authors' research.

The focus groups conducted in this qualitative study were based on a semi-structured interview guide, which included only open-ended questions specific to this type of research, intended to elicit opinions and views from participants [84]. The questions were designed and structured around five themes based on the literature (Table 2).

**Table 2.** Design of the focus group questions based on the literature.

| Question Asking | Issue: Why is This Being Asked? | Author(s) |
|---|---|---|
| **Theme 1: Factors influencing the vacation purchase decision**<br>Q1. What are the primary factors that influence your decision-making process when purchasing a holiday?<br>Q2. What factors do you take into consideration when selecting accommodation for your holiday? | to explore the main elements that influence Generation Z's decision-making process when purchasing vacations | Setiawan et al. (2018) [21] |
| **Theme 2. Sources of travel information**<br>Q3. What sources do you usually rely on to find information about your travel plans?<br>Q4. From where do you seek inspiration when selecting your vacation destinations? | to investigate the main sources of information used by Generation Z regarding travel | Reisenwitz and Fowler (2015) [85]<br>Dimitriou and AbouElgheit (2019) [40] |
| **Theme 3. Distribution channels used for reservations**<br>Q5. What methods do you typically use to make your holiday bookings?<br>Q6. What factors have influenced your decision-making process in selecting your preferred booking method thus far? | to identify the distribution channels used by Generation Z to purchase vacations | Prayag et al. (2014) [86]<br>Jedin et al. (2020) [87]<br>Perčić et al. (2021) [24] |
| **Theme 4. The use of social media during the travel**<br>Q6. Do you engage in the practice of sharing photos and videos from your vacations on social media platforms?<br>Q7. If you do share photos and videos from your vacations on social media, what are the primary motivations behind this practice? | to evaluate the importance of social media during travel for Generation Z | Lalicic et al. (2016) [61]<br>Jansson et al. (2018) [62]<br>Xiang et al. (2010) [88]<br>Chung et al. (2015) [89]<br>Dimitriou and AbouElgheit (2019) [40] |
| **Theme 5. The influence of advanced technologies on travel**<br>Q8. Have you ever utilized artificial intelligence to discover holiday ideas?<br>Q9. What is your perspective on the substitution of human resources in the tourism industry with artificial intelligence?<br>Q10. Do you anticipate that virtual reality will impact your future travel experiences? Please provide a justification for your response. | to assess the extent to which advanced technologies may influence the future Travel behavior of Generation Z. | Buhalis and Karatay (2022) [26]<br>Buhalis et al. (2023) [49]<br>Dwivedi et al. (2023) [79] |

Source: the authors' research.

The discussions were moderated by one of the article's authors and took place in early March, with each interview lasting approximately 1 h and 20 min. Participants were informed about the location, time, and general theme of the research one week prior to the interview.

The two group interview sessions started by emphasizing the rules of good practice so that all participants feel comfortable and free to express their viewpoints. Participants were duly informed that the discussions would be subject to audio recording to facilitate accurate transcription. Their explicit consent was sought exclusively for research purposes pertaining to the study. Furthermore, meticulous attention was given to outlining the pertinent aspects of confidentiality.

Throughout the discussions, the moderator's role was limited to guiding the conversations, refraining from making any personal judgments that could potentially sway the participants' perspectives. Participants were actively encouraged to engage in the

discussions and freely express their beliefs without any limitations, with the assurance that their opinions and experiences held equal significance in fulfilling the research objectives.

The audio-recorded discussions were transcribed and analyzed using the content analysis method [90]. The transcript of the discussions underwent analysis to identify the most pertinent content aligned with the research objectives. The content analysis process was initiated by undertaking a thorough familiarization stage, involving attentive listening to the recorded discussions and careful reading of the corresponding transcripts [91]. Next, a meticulous analysis of each speech was conducted to identify the principal topics in depth. Finally, the information provided by the participants was systematically coded and standardized in alignment with the research objectives. Expressions conveying similar meanings, regardless of variations in linguistic expression or temporal occurrence, were included in the same response category. The results of the coding process have been incorporated into a comprehensive content analysis table, detailed in Appendices A and B.

## 4. Results

The focus group discussion commenced with the researcher requesting participants to provide a brief account of their most recent vacation experience, outlining the main elements that led to their choice. The main objective of this introductory question was to encourage participants to share their personal experiences without hesitation or fear of judgment. By creating an environment where participants felt comfortable and confident in sharing their experiences, the aim was to encourage dynamic interactions between group members. This approach sought to stimulate engaging discussions highlighting commonalities and differences in their experiences.

From this introductory part, the authors created a short tourism profile of the members of Generation Z analyzed (Table 3). They prefer and use the airplane as a transportation means, followed by the personal car. The main reasons for travel mentioned were exploring cities, getting to know local communities, discovering new cultures, and relaxing. City breaks, seaside vacations, and exotic trips were mentioned. Half of the respondents prefer short holidays, between three and five days, and the other half prefer longer holidays, from one week to two weeks.

**Table 3.** Tourism profile of the members of Generation Z.

| Transport | Main Reasons | Type of Vacation | Duration |
|---|---|---|---|
| Plane<br>Car | Exploring cities<br>Getting to know local communities<br>Discovering new cultures<br>Relaxing | City Breaks<br>Seaside vacations<br>Exotic trips | 3–5 days<br>7–14 days |

Source: the authors' research.

### 4.1. Factors That Influence the Holiday Purchase Decision

The main factor behind the decision to buy a holiday, mentioned by 17 of the 20 respondents, is price. Elements such as the season, tourist attractions, and activities that can be done at the destination were also mentioned with much lower frequency.

*"Tourist attractions are very important to me; I do not go to a place if there is nothing to do there. Of course, the price matters, but more important is to improve my general knowledge after a tourist experience."* (Subject 5)

A possible event in a certain tourist destination was brought up as a decision factor:

*"I chose a tourist destination for next summer because of a concert of one of my favorite bands."* (Subject 5)

Regarding the factors they consider when choosing an accommodation, they also mentioned price, followed in order of frequency of responses, location, facilities, and reviews.

> *"It also matters a lot how much time you spend at the destination, for example if I go on holiday for 3 days the location matters a lot, if I stay longer, other aspects matter more, such as price or facilities. Reviews are also important."* (Subject 2)

*4.2. Travel Information Sources*

Members of Generation Z unanimously mention the internet as their primary source of information when it comes to travel. Additionally, 50% of the participants nominated social media networks, followed equally by travel agencies and relatives/friends.

> *"My main sources of information about holidays are relatives, friends, social media, especially TikTok"* (Subject 1)

Only one participant mentioned artificial intelligence as a means of searching for information about tourist destinations:

> *"I used the GPT Chat platform for information on certain destinations . . . it gave me some super cool recommendations, like talking to a local in the area . . . I searched for answers to questions like <What to visit in city X> and <How many days are enough to visit destination X.>"* (Subject 11)

The information provided by this intelligent platform seemed relevant to Subject 11, who also said that this technology can save a lot of time:

> *"I don't have to spend time reading five blogs to find out one piece of information."* (Subject 11)

In response to the reports presented in subject 11, subject 17 interjected in the discourse, presenting a compelling argument regarding the efficacy of this intelligent application within the realm of tourism.

> *"If you do a Google search for a holiday in the Maldives, for example, 100 web pages come up, but Chat GPT combines them all and summarises the relevant information on one page in seconds"* (Subject 17)

As sources of inspiration, the following were mentioned in ascending order: influencers (travel bloggers) on social media, vacation stories from friends, posts from virtual friends on social networks, newsletters received by e-mail from the travel agencies where they subscribed, as well as content created by travel agencies on social media.

> *"When I see pictures or videos in the stories of influencers I follow, and I immediately want to visit those places . . . they come with real information, you see it in black and white, not like on TV where they cover everything up."* (Subject 18)

> *"I actually saw a video on TikTok, and I was curious to look for my plane ticket to that destination"* (Subject 8)

> *"I find it useful to follow different travel bloggers on social networks because they travel a lot and take part in many activities and so the decision to choose a certain destination can become easier."* (Subject 17)

> *"I follow travel influencers, but so far I haven't been to any of their recommended destinations."* (Subject 20)

> *"I follow the social media pages of travel agencies, even they have started making their own reels."* (Subject 1)

While social media influencers were frequently cited as a significant source of inspiration for potential future trips, there were still respondents who expressed differing opinions and challenged the prevailing views.

> *"I do not trust them, neither do I follow such people in social media, I think they get paid for it, so they must sell well. I tend to go more for online reviews and recommendations from friends."* (Subject 12)

### 4.3. Distribution Channels Used for Making Reservations

The respondents were familiar with direct and indirect distribution channels for selling tourism products and services. They choose how to make their bookings depending on the specific type of vacation. For example, for trips to distant destinations, they prefer to use traditional travel agencies, while for more easily accessible destinations, they opt for online travel platforms.

*"I usually make my travel reservations through travel agencies, or on Booking.com, or directly on accommodation websites."* (Subject 5)

*"When I want all-inclusive holidays I use travel agencies, otherwise I use Booking.com and airline websites."* (Subject 13)

*"I think about the destination first, then I analyse the market a bit. For example, this year I bought a holiday from a travel agency. I find out what they can offer to me, flight, transfer, accommodation, but there have been times when I have booked directly on the hotel website, directly on the airline's website . . . if it is a more far away destination, where you need more information, visa and so on, it is better with an agency, if it's an easily accessible destination you can do your own bookings."* (Subject 2)

The online travel agency Booking.com was the most mentioned travel platform and the reasons respondents chose to use it were its popularity, ease of use, and multi-functionality.

*"I prefer Booking.com because it integrates both the booking and the recommendation side in one place."* (Subject 18)

Among those who prefer to purchase tourist services directly from the websites of accommodation establishments and transport companies, the reasons for their choice were the avoidance of intermediaries' commissions and their distrust of them.

*"First, I look on Booking.com, then I check the hotel website, because I do not trust Booking completely. Some people I know have had problems with booking through this platform."* (Subject 14)

Reasons for choosing traditional travel agencies were highlighted as reducing search time, the possibility of purchasing multiple tourism services in a package from a single location, and safety.

*"However, so far, the most okay was when I purchased a package from a travel agency with all services included. That way I saved time and the prices were better."* (Subject 2)

*"I prefer travel agencies for holidays in faraway destinations because if something happens you have someone to call."* (Subject 1)

### 4.4. The Use of Social Media during Travels

Regarding using social media while traveling, the opinions of Generation Z were divided. More than half of the respondents always share photos on social media and some of them only occasionally. The main reasons for the first category mentioned above are: sharing pictures with friends in cyberspace and potentially influencing them, keeping memories, and social proof. A proportion of respondents said they do not do this for a specific reason, but just for pleasure.

*"I post on Instagram, Facebook, TikTok so people can see where I am, and maybe I can influence my friends to visit those places too."* (Subject 3)

*"I post on Facebook and Instagram. On average I spend 4 h on my phone on a three-day vacation, and in those hours, I post a lot. In general, I post to raise my social profile."* (Subject 12)

There were opinions among respondents that they prefer to post photos rather when they return from vacation. Others mentioned that they do not post photos at all, considering their vacations to be personal issues.

*"I do not like to post; I consider it my experience and keep it to myself."* (Subject 14)

*"I don't really post from holidays; I like to have my time and experience the destination."* (Subject 17)

All respondents who reported sharing content while traveling exclusively mentioned utilizing Instagram as their platform of choice for doing so. Facebook (4 times), TikTok (1 time), and Snapchat (1 time) were also mentioned.

*4.5. The Influence of Advanced Technologies on the Future of Travel*

Of the 20 respondents, only one had used artificial intelligence to search for ideas and information related to tourist destinations, as reported above in the section related to sources of information when it comes to travel.

Regarding the possibility of replacing human resources in tourism with artificial intelligence, they believe this will happen, but not soon. They consider that although their generation is more adaptable, there are still older generations who are not ready for such a leap. Most estimated between five and 10 years before artificial intelligence is used on a wider scale.

*"I believe that it will take at least 10 years to adapt to them, our generation is not ready for such a leap, maybe the Alpha generation, they were born with them, we have taken them by the hand . . . I think there is the necessary technology, the necessary evolution in the immediate period, but we humans are not ready."* (Subject 12)

*"I believe that just as in other fields humans have been replaced by robots (e.g., banking), the same will happen with tourism specialists, but this is more complicated and will take a long time because there are still older generations that cannot adapt so easily to artificial intelligence."* (Subject 2)

*"I believe that it will take many more years for these technologies to penetrate and take over the mass of the population, as there are older generations who would find it difficult to adapt.")* (Subject 11)

Some of the participants expressed their views on the relationship between artificial intelligence and human resilience as follows:

*"It would be interesting if there was a robot that could figure out how you feel and what you want, so it could recommend exactly what you desire, but I still think that behind the robots there will be people controlling them."* (Subject 5)

*"Behind a robot will still be a human being."* (Subject 3)

To the question about using virtual reality to explore a destination or a tourist attraction, only one of the participants mentioned that he had visited a museum in virtual reality and was delighted with the experience, but would limit the experience to that level only, preferring to visit destinations physically, so that he could experience with his senses everything about a location (people, culture, traditions).

Regarding the possible impact of advanced technologies, such as mixed reality and the Metaverse, on future travel methods, they unanimously answered that exclusively virtual travel options are not being considered.

*"We are human, we have feelings, I can visualize a location, but I can't tan through VR goggles . . . you don't feel like you're on vacation, you'd be in some kind of game, I can imagine what a pleasure it would be to ski from a joystick."* (Subject 18)

*"The most this technology can do for travel is make you curious to want to physically go to that location."* (Subject 3)

However, half of the participants specified that these technologies would be extremely useful in the planning stage of their travels.

*"The trip itself will still be a classic one, but it would be interesting to see what's at your destination before you hit the road."* (Subject 1)

Although, as mentioned above, participants excluded the possibility of virtual travel in the future, one respondent did have an intervention on this issue:

*"Although I like to experience destinations with my own senses, I like the idea of virtually visiting dangerous destinations, or where there are prohibitions. It intrigues me that I cannot get there, and the Metaverse might be the solution."* (Subject 11)

To facilitate the understanding of the research results, the authors opted for a visual presentation of the most significant findings through a visual tool (Figure 1).

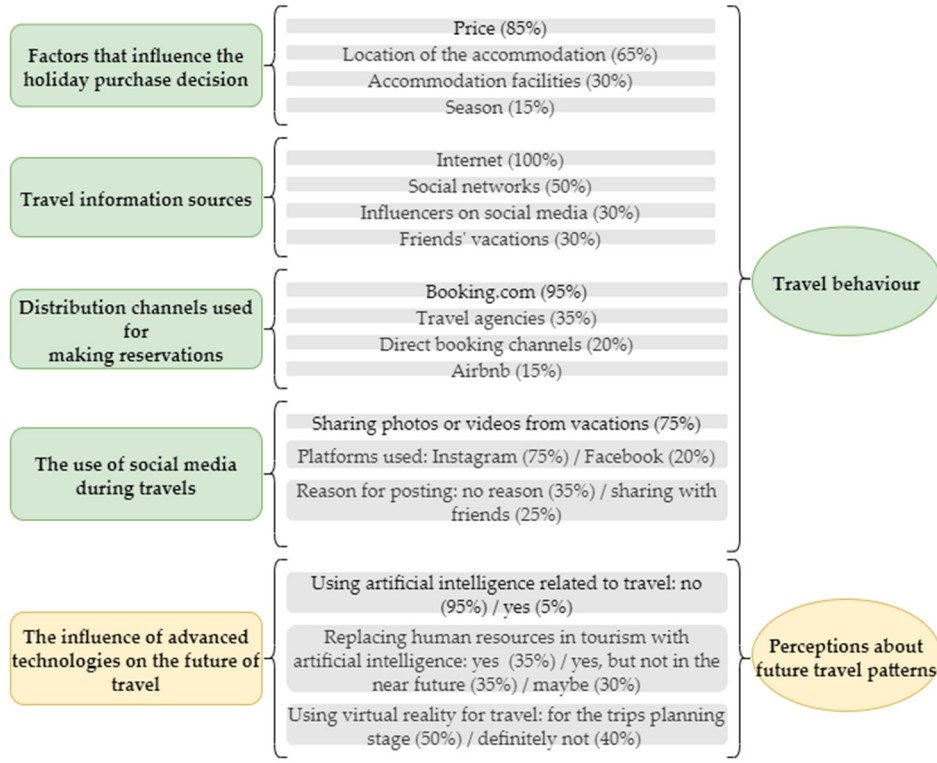

**Figure 1.** Main research results. Source: authors' research.

The research findings indicate that the disruptive behavior of Generation Z tourists can be largely attributed to their utilization of social media platforms, particularly for travel purposes. Social media has emerged as a dominant search engine for this demographic, and travel influencers significantly influence certain Generation Z individuals. In addition, they also act as influencers, with most of them posting photo or video content during the trip. Another notable aspect within the travel industry is the propensity of Generation Z tourists to employ multi-channel booking, seamlessly transitioning between various booking options with minimal effort. Lastly, the willingness of Generation Z tourists to embrace advanced technologies such as virtual reality during trip planning presents an additional factor that can disrupt conventional travel patterns.

## 5. Discussion

This paper has attempted to fill a gap in the literature regarding the behavior of Generation Z tourists in Romania and their perception of the impact that advanced technologies may have on the future of travel.

Several similarities were identified in relation to the behaviors of this generation captured in the literature, as well as specific characteristics of the group under analysis. According to an Amazon Ads study [92], Generation Z is more likely to channel their desire to plan trips to destinations where they can relax on the beach and get lost in big cities, reflecting the broader trend of longer-distance travel facilitated by air travel.

Another interesting aspect of this research is related to the fact that, although none of the respondents brought up their concern for sustainability in a conscious or perceived way, some of the respondents emphasized the discovery of local people and their culture, contrary to Haddouche and Salomone's study [16] done among Generation Z in France, which states that the discovery of "others" was not a current theme among the respondents they analyzed.

The results of this research show that price is an important consideration in their decisions to buy tourist products and services, as also revealed in the study by Subawa et al. [93]. This could be due to the multitude of "low-cost" companies in the tourist market and the various websites that "teach" you how to travel on a low budget. Thus, they may feel tricked into buying services at a higher price than other people do; hence, their concern is to find the best deals.

Interesting was the debate related to social media travel influencers. Although digital natives, and tech-savvy, not all respondents consider social media influencers credible sources of inspiration.

In terms of how they purchase their holidays, Generation Z members surveyed are familiar with all distribution channels, from direct to intermediaries. They demonstrated that they know the advantages of using each and can easily navigate and switch from one channel to another. They can also use multiple channels for a single vacation. The conventional travel agency was mentioned more than once in the discussion with Generation Z surveyed, because of the ease of purchasing a package with several services included, or a holiday in a faraway destination. Booking.com is used both for booking and viewing reviews that contribute to their final decision to buy a holiday. Sharing economy platforms, such as Airbnb, were mentioned by respondents but in a lower percentage than Booking.com. The preference for the use of online travel agencies regardless of the age of the tourists was also demonstrated by Jedin et al. study [87].

Specific to Generation Z in terms of tourist behavior is that they post photos and videos of their vacations on social media platforms, thus becoming influencers for their virtual friends. The importance of social media among Generation Z tourists has also been highlighted in Xiang et al. [88] and Chung et al. [88].

In terms of Generation Z's perception of the influence of advanced technologies, especially virtual reality, on future travel, the survey results illustrate, as in Buhalis and Karatay [26], that respondents are not aware of the full capabilities of these technologies. This confirms what Kotler et al. states in the book Marketing 5.0 [94] (p.76) that in tourism "digitalization is mostly on the surface level and not yet at a transformative level". This idea is reinforced by the results of this research on artificial intelligence and the robotization of industry. Although these new tools help in the planning phase, they do not exclude human resources as facilitators of tourism experiences.

## 6. Conclusions and Implications

By disseminating these findings, this research contributes academically to enhancing a better understanding of the behavioral patterns exhibited by Generation Z tourists in the context of technological developments. It specifically emphasizes key elements that possess the potential to disrupt the industry, thereby offering valuable insights for stakeholders seeking to comprehend and adapt to the preferences and tendencies of this demographic group.

The responses to the first survey question regarding how Generation Z organizes their vacations shed light on elements that hold disruptive potential for traditional approaches to travel organization. These include the utilization of social media and influencers as sources of travel search and inspiration, challenging conventional methods.

Regarding the second research question, which explores the use and significance of social media platforms during travel, a noteworthy behavior emerges: Generation Z individuals themselves become influencers by sharing photo and video content, thus influencing others in their social networks.

The third research question investigates Generation Z's perception of the future of tourism in the context of technological advancements. It reveals their limited awareness of the potential of certain technologies, yet they exhibit a willingness to adopt those that can simplify the travel planning process, such as artificial intelligence and virtual reality.

In conclusion, the answers to the research questions brought to light valuable elements regarding the potential of the emerging generation of tourists.

This study adds new knowledge to the literature on the relationship between the Romanian Generation Z and their tourism consumption behaviors, as well as their views on the impact of advanced technologies on the industry.

The findings of this study also offer practical implications for tourism business managers and marketers in the tourism industry. By identifying the behavior of digital tourists, they can create appropriate marketing strategies for targeting and attracting this segment of travelers. For instance, tourism business managers could attract this category of consumers by focusing on creating personalized tourism products and services that make them feel unique. Managers should also recognize the significance of utilizing multiple distribution channels, given the ease with which young people utilize these channels. Marketers in the industry could increase their use of Generation Z's favorite social media platforms, such as Instagram and TikTok, by developing effective content strategies to engage with them. Furthermore, marketing agents, together with travel business managers, could streamline the travel destination selection process for Generation Z tourists by gradually introducing advanced technologies such as artificial intelligence and virtual reality.

The findings of this study approach interesting debates about the disruptive potential of digital natives in the tourism industry; however, several limitations of this study should be considered, which offer perspectives for future research. Firstly, a limitation of this research is the age category of participants, which does not include all segments of Generation Z. Secondly, participants with different levels of education were not included, only students from the same University. Thirdly, another limitation is given by the qualitative nature of the research, which usually involves a small number of participants. Although this study does not pretend to extrapolate the results to the macro level, it can be considered a main starting point for setting objectives and hypotheses for a future more complex quantitative research with a larger number of participants. Additionally, possible future research could be to explore the behaviors of Generation Z tourists in several countries and compare them, potentially providing insights into how national culture and other factors influence the behavior of young tourists. An interesting way forward for future research could involve conducting an experiential study in which members of Generation Z use virtual reality glasses to explore a destination. They could then provide immediate feedback and impressions following the virtual experience. This approach could provide valuable insights into the impact of virtual reality on the perception and evaluation of travel destinations among this demographic group.

**Author Contributions:** Conceptualization, A.P.P.V., C.A.B., G.B., A.S.T., I.B.C. and L.D.; methodology, A.P.P.V., C.A.B., G.B., A.S.T., I.B.C. and L.D.; literature review, A.P.P.V., C.A.B., G.B., A.S.T., I.B.C. and L.D.; writing—original draft preparation, A.P.P.V., C.A.B., G.B, A.S.T., I.B.C. and L.D.; writing—review and editing, C.A.B.; visualization, A.P.P.V., C.A.B., G.B, A.S.T., I.B.C. and L.D.; supervision, C.A.B.; project administration, C.A.B.; funding acquisition, C.A.B. All authors have read and agreed to the published version of the manuscript.

**Funding:** This research was funded by Transilvania University of Brașov, Romania.

**Institutional Review Board Statement:** Ethics review and approval was waived due to the non-interventional nature of the study. However, all subjects were informed about the study and participation was entirely voluntary.

**Informed Consent Statement:** Not applicable.

**Data Availability Statement:** The data presented in this study are available on request from the corresponding author.

**Conflicts of Interest:** The authors declare no conflict of interest.

## Appendix A

**Table A1.** Horizontal content analysis, first group interview session.

| Themes | S1 | S2 | S3 | S4 | S5 | S6 | S7 | S8 | S9 | S10 | HA |
|---|---|---|---|---|---|---|---|---|---|---|---|
| **Influences on holiday purchase decision** | | | | | | | | | | | |
| **Main factors** | 1.price | 1. price 2. season | 1. price 3. friends | 1. price | 4. sights | 2. season | 1. price | 1. price | 1. price | 1.price | 1-8× 2-2× 3-1× 4-1× |
| **Accommodation** | 1. facilitates 2. location | 2. location 3. reviews. | 1. facilitates 4. price | 4. price | 1. facilitates 2. location 3. reviews | 1. price 2. location | 3. reviews | 3. reviews 4. price | 2. location 4. price | 2. location 4. price | 1-3× 2-6× 3-4× 4-5× |
| **Travel information sources** | | | | | | | | | | | |
| **Main sources of information** | 1. internet 2. social networks | 1. internet 3. travel agencies 4. Google Ads | 1. internet 2. social networks 5. friends | 1. internet | 1. internet 5. friends | 1. internet 2. social networks | 1. internet 2. social networks 3. travel agencies | 1. internet 2. social networks 5. friends | 1. internet 2. social networks | 1. internet | 1-10× 2-6× 3-2× 4-1× 5-3× |
| **Main sources of inspiration** | 1. friends' social media posts 2. travel agency social media posts | 3. newsletters from travel agencies | 1. friends' social media posts | 4. friends' vacations | 4. friends' vacations | 1. friends' social media posts | 3. newsletters from travel agencies 5. Influencers 6. travel agencies' posters | 5. Influencers | 4. friends' vacations | 1. friends' social media posts | 1-4× 2-1× 3-2× 4-3× 5-2× 6-1× |
| **Distribution channels** | | | | | | | | | | | |
| **Booking processing** | 1. travel agencies | 1. travel agencies 2. Booking.com 3. direct bookings | 1. travel agencies 2. Booking.com | 2. Booking.com | 1. travel agencies 2. Booking.com 3. direct bookings | 2. Booking.com | 1. travel agencies 2. Booking.com | 2. Booking.com | 2. Booking.com | 2. Booking.com | 1-5× 2-9× 3-2× |
| **Reason for the choice of booking methods** | 1. reducing search time | 1. reducing search time 2.buying a package from one place | 1. reducing search time | 3. ease of use | 1. reducing search time 3. ease of use 4. avoiding commission | 3. ease of use | 1. reducing search time 3. ease of use | 3. ease of use | 3. ease of use | 3. ease of use | 1-5× 2-1× 3-6× 4-1× |
| **Using socializing platforms while traveling** | | | | | | | | | | | |
| **Sharing holiday photos and videos on social networks** | 1. yes, always | 1. yes, always | 1. yes, always | 1. yes, always | 1. yes, always | 2. occasionally | 1. yes, always | 1. yes, always | 2. occasionally | 3. no | 1-7× 2-2× 3-1× |
| **Networks where they share holiday photos** | 1. Instagram | 1. Instagram | 1. Instagram 2. Facebook 3. TikTok | 1. Instagram | 1. Instagram 2. Facebook | 1. Instagram 4. Snapchat | 1. Instagram | 1. Instagram | 1. Instagram | - | 1-9× 2-2× 3-1× 4-1× |
| **Reason for sharing holiday photos** | 1.keeping memories | 1.keeping memories | 2. social proof 5. sharing with friends | 4. for no reason | 5. sharing with friends | 5. sharing with friends | 4. for no reason | 4. for no reason | 4. for no reason | - | 1-2× 2-1× 3-1× 4-4× 5-3× |

**Table A1.** *Cont.*

| Themes | S1 | S2 | S3 | S4 | S5 | S6 | S7 | S8 | S9 | S10 | HA |
|---|---|---|---|---|---|---|---|---|---|---|---|
| **Influence of advanced technologies** | | | | | | | | | | | |
| **Using artificial intelligence to find travel-related information** | 1. no | 1. no | 1. no | 1. no | 1. no | 1. no | 1. no | 1. no | 1. no | 1. no | 1-10× |
| **Replacing human resources in tourism with artificial intelligence** | 1. yes, but not in the near future | 1. yes, but not in the near future | 2. maybe | 1. yes, but not in the near future | 2. maybe | 2. maybe | 2. maybe | 1. yes, but not in the near future | 3. yes | 3. yes | 1-4× 2-4× 3-2× |
| **Using virtual reality for travel** | 1. only at the trip planning stage | 2. no | 2. no | 1. only at the trip planning stage | 1. only at the trip planning stage | 2. no | 2. no | 1. only at the trip planning stage | 3. only for sightseeing | 1. only at the trip planning stage | 1-5× 2-4× 3-1× |

Notes: HA–Horizontal analysis; S–subject; ×–frequency of occurrences (e.g., 2× means twice).

## Appendix B

**Table A2.** Horizontal content analysis, the second group interview session.

| Themes | S11 | S12 | S13 | S14 | S15 | S16 | S17 | S18 | S19 | S20 | *HA* |
|---|---|---|---|---|---|---|---|---|---|---|---|
| **Influences on holiday purchase decision** | | | | | | | | | | | |
| **Main factors** | 1.price 2. season | 1. price | 1. price 3. friends | 1. price | 1. price | 1. price | 5. activities that can be done at destination | 1. price | 1. price 5. activities that can be done at destination | 1.price 4. sights | 1-9× 2-1× 3-1× 4-1× 5-2× |
| **Accommodation** | 1. facilitates | 3. reviews | 1. facilitates 4. price | 4. price | 1. facilitates 2. location | 4. price | 4. price | 2. location | 4. price | 2. location 4. price | 1-3× 2-3× 3-1× 4-6× |
| **Travel information sources** | | | | | | | | | | | |
| **Main sources of information** | 1. internet 2. social networks 6. artificial intelligence | 1. internet 5. friends | 1. internet 3. travel agencies | 1. internet 3. travel agencies | 1. internet | 1. internet | 1. internet 2. social networks | 1. internet 2. social networks | 1. internet | 1. internet 2. social networks | 1-10× 2-4× 3-2× 5-1× 6-1× |
| **Main sources of inspiration** | 1. friends' social media posts 5. Influencers | 4. friends' vacations | 4. friends' vacations 5. Influencers | 7. it's not inspired from anywhere | 4. friends' vacations | 7. it's not inspired from anywhere | 5. Influencers | 5. Influencers | 7. it's not inspired from anywhere | 5. Influencers | 1-1× 4- 3× 5-4× 7-3× |
| **Distribution channels** | | | | | | | | | | | |
| **Booking processing** | 2. Booking.com 4. Airbnb | 2. Booking.com | 1. travel agencies 2. Booking.com 3. direct bookings | 1. travel agencies 2. Booking.com 3. direct bookings | 2. Booking.com 4. Airbnb | 2. Booking.com | 2. Booking.com | 2. Booking.com | 2. Booking.com | 2. Booking.com 4. Airbnb | 1-2× 2-10× 3-2× 4-3× |
| **Reason for the choice of booking methods** | 1. reducing search time | 1. reducing search time | 1. reducing search time 2. buying a package from one place | 4. avoiding commission | 5. popularity of online platforms | 3. ease of use 5. popularity of online platforms | 3. ease of use | 3. ease of use | 3. ease of use 5. popularity of online platforms | 3. ease of use 5. popularity of online platforms | 1-3× 2-1× 3-5× 4-1× 5-4× |

**Table A2.** *Cont.*

| Themes | S11 | S12 | S13 | S14 | S15 | S16 | S17 | S18 | S19 | S20 | *HA* |
|---|---|---|---|---|---|---|---|---|---|---|---|
| **Using socializing platforms while traveling** | | | | | | | | | | | |
| **Sharing holiday photos on social networks** | 1.yes, always | 1. yes, always | 1. yes, always | 3. no | 2. occasionally | 3. no | 3. no | 1. yes, always | 2. occasionally | 3. no | 1-4× 2-2× 3-4× |
| **Networks where they share holiday photos** | 1. Instagram 2. Facebook | 1. Instagram 2. Facebook | 1. Instagram | - | 1. Instagram | - | - | 1. Instagram | 1. Instagram | - | 1-6× 2-2× |
| **Reason for sharing holiday photos** | 5. sharing with friends 6. socializing | 2. social proof | 4. for no reason | - | 4. for no reason | - | - | 5. sharing with friends | 4. for no reason | - | 2-1× 4-3× 5-2× 6-1× |
| **Influence of advanced technologies** | | | | | | | | | | | |
| **Using artificial intelligence to find travel-related information** | 2. yes | 1. no | 1. no | 1. no | 1. no | 1. no | 1. no | 1. no | 1. no | 1. no | 1-9× 2-1× |
| **Replacing human resources in tourism with artificial intelligence** | 1. yes, but not in the near future | 1. yes, but not in the near future | 3. yes | 1. yes, but not in the near future | 2. maybe | 3. yes | 3.yes | 3. yes | 3. yes | 2. maybe | 1-3× 2-2× 3-5× |
| **Using virtual reality for travel** | 4. only for visiting more inaccessible destinations | 2. no | 1. only at the trip planning stage | 1. only at the trip planning stage | 2. no | 2. no | 1. only at the trip planning stage | 2. no | 1. only at the trip planning stage | 1. only at the trip planning stage | 1-5× 2-4× 4-1× |

Notes: HA–Horizontal analysis; S–subject; ×–frequency of occurrences (e.g., 2× means twice).

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
