# Peer review of "Examining the Disruptive Potential of Generation Z Tourists on the Travel Industry in the Digital Age"

_sustainability, doi:10.3390/su15118756_

Round 1
Reviewer 1 Report
This is an interesting research topic and the author use focus group method is appropriate. I suggest the authors improve their writing ability on both theory and methodology, and I provide some suggestions for this study to improve the following aspects:
1. Without information on the demographic characteristics of the respondents, it is impossible to determine whether the representative sample of Generation Z is limited by cultural background and nationality.
2. There is no sufficient discussion of theoretical research gap, and in the discussion section there is no theoretical contributions and further research field.
3. This study does not demonstrate the procedural and academic rigor of the focused group, including: 1) interview question design; 2) control and guidance of the discussion process; and 3) the coding process only includes some descriptions, and data analysis results lack of group dynamic and interaction.
Author Response
Dear Reviewer #1,
please find attached our answers.
Thank you!

Reviewer 2 Report
The topic covered is important and it is essential to have analyzes carried out in different contexts.
The use of focus groups is appropriate for the objectives of the study and the authors are aware of the main limitations, well described in the last paragraph.
I agree with what reported by the authors and in particular the need for "a future more complex quantitative research with a larger number of participants". In any case, even with the recognized limits, some improvements can be made.
In reality, the paper reveals a profile of the interviewees that is not so innovative after all, with many traits of behavior that I would define as "wise" and very close to those of previous generations.
I therefore suggest that you look into some aspects:
- verify in the literature studies that compare the behavior of generation Z with the previous ones
- better structure the description of the profile which in some passages seems to be presented as already "disruptive", while in the analysis of the results it appears very "normal"
- better focus and make the potential disruptive effects more evident, to avoid them appearing, as described in the paper, as slogans rather than the results of a scientific work.
Author Response
Dear Reviewer #2,
please find attached our answers.
Thank you!

Reviewer 3 Report
Dear Authors!
Let me start by saying that «Examining the disruptive potential of Generation Z tourists on the travel industry in the digital age» is a very interesting topic and context.
Thank you for the possibility review this manuscript. I am sympathetic with the article, but I envisage some changes needed before publication.
- It is advisable to finalize the abstract. The abstract must include sufficient information for readers to judge the nature and significance of the topic. The abstract should contain the main idea of the paper, the subject and the goal of the research, methods used, hypotheses, research results and a brief conclusion.
- Perhaps it is better not to ask questions, but to formulate hypotheses.
- The text of the article should be structured. The literature review should also be concluded with 2-3 summarizing sentences. Then the purpose of the research should be formulated. After that, you should state (if you foresee it) hypotheses. It is necessary to provide a part of the material obtained as a result of the focus groups in graphical form using relative values.
- The Conclusions paragraph is missing. The conclusions should be finalized. Conclusions should be reinforced with analytical data based on research results.
I am grateful to the authors for the interesting material they have prepared.
Best regards
Author Response
Dear Reviewer #3,
please find attached our answers.
Thank you!

Round 2
Reviewer 2 Report
The authors satisfactorily integrated and improved the paper according to the comments and requests.
Author Response
Dear Reviewer #2,
thank you for your appreciation and once again for your involvement which has led to the improvement of the substatial work.
Sincerely,
Authors